# Systemic and respiratory T-cells induced by seasonal H1N1 influenza protect against pandemic H2N2 in ferrets

Koen van de Ven [1], Femke de Heij[1,3], Harry van Dijken[1], José A. Ferreira[2] & Jørgen de Jonge [1✉]

Traditional influenza vaccines primarily induce a narrow antibody response that offers no protection against heterosubtypic infections. Murine studies have shown that T cells can protect against a broad range of influenza strains. However, ferrets are a more potent model for studying immune correlates of protection in influenza infection. We therefore set out to investigate the role of systemic and respiratory T cells in the protection against hetero-subtypic influenza A infections in ferrets. H1N1-priming induced systemic and respiratory T cells that responded against pandemic H2N2 and correlated with reduced viral replication and disease. CD8-positive T cell responses in the upper and lower respiratory tract were exceptionally high. We additionally confirmed that H2N2-responsive T cells are present in healthy human blood donors. These findings underline the importance of the T cell response in influenza immunity and show that T cells are a potent target for future universal influenza vaccines.

[1] Centre for Infectious Disease Control, National Institute for Public Health and the Environment (RIVM), Bilthoven, the Netherlands. [2] Department of Statistics, Informatics and Modelling, National Institute for Public Health and the Environment (RIVM), Bilthoven, the Netherlands. [3]Present address: Princess Máxima Center for Pediatric Oncology, Utrecht, the Netherlands. ✉email: jorgen.de.jonge@rivm.nl

          **1**

nfluenza A virus (IAV) remains a threat to human health despite the availability of vaccines[1]. Traditional influenza vaccines mainly activate the humoral immune response against the surface proteins haemagglutinin (HA) and neuraminidase (NA; reviewed in Keshavarz et al.[2]). However, HA and NA can mutate over time (antigenic drift) and multiple influenza strains can reassort to establish a new virus with altered HA and/ or NA composition (antigenic shift; reviewed in Taubenberger and Kash[3]). Both antigenic drift and shift can lead to evasion of the antibody response. In contrast to surface proteins, the internal influenza proteins are far less subject to change and are highly conserved between influenza subtypes. Epitopes of internal IAV proteins can be recognized by T cells that can either kill virus-infected cells or assist in the development of adaptive immune responses (reviewed in refs. [4,5]).

The impact of influenza strains evading the human immune response is illustrated by the pandemic of H2N2 in 1957. After introduction into the human population the virus spread rapidly, leading to ~1–2 million deaths worldwide[6,7]. Although H2N2 disappeared from the human population in 1968, there is a risk of reintroduction as associated-H2N2 strains still circulate in birds[8,9]. Moreover, persons born after 1968 have not encountered H2N2 and will likely have no neutralizing antibodies against H2N2[7,10]. With the pandemic history of H2N2, a possible reemergence in humans should be considered.

T cells might offer protection if H2N2 reemerges. The Cleveland Family study already reported in 1958 that prior exposure to H1N1 influenza resulted in reduced disease in participants infected with H2N2[11,12]. This was independent of neutralizing antibodies, hinting toward a role for T cells. Later studies provided more evidence that T cells can protect against heterosubtypic IAV infections. Pre-existing influenza-specific CD8 T cells correlated with decreased disease burden during the H1N1 pandemic of 2009[13] and were also found to contribute to recovery from H7N9 infection[14]. In a human challenge study, pre-existing CD4 T cells responding to internal influenza proteins were associated with reduced virus shedding and disease upon infection with seasonal H1N1 and H3N2[15].

Animal studies provided additional proof that T cells are essential for heterosubtypic immunity. In ferrets, prior exposure to influenza protected against heterosubtypic infections with H1N1[16], H3N2[17], or H5N1[18], which was likely mediated by T cells. Murine research demonstrated that T cells in general (reviewed in Altenburg et al.[19]) – and especially tissue-resident memory T cells (Trm)[20–23] – are crucial for protection against heterosubtypic IAV infections. Trm is a population of non-circulating T cells that is located near the site of infection and can respond rapidly upon (recurring) infections with the production of cytokines and killing of infected cells (reviewed in Rosato et al.[24]). Trm have also been identified in human lungs[25–29], but the availability of tissue and ethical concerns are hindering a more in-depth investigation.

The rol of T cells in influenza infections is often studied in murine models, which offer a wide variety of techniques and reagents but are limited as an influenza model. Mice do not display the traditional disease symptoms (fever, sneezing, etc.) of an influenza infection and virus strains often require adaptation to increase virulence in mice[30]. Hence extrapolation to humans is difficult and it is unknown how well the findings from murine studies translate to humans. In contrast, ferrets show typical symptoms of influenza disease and are susceptible to both avian and human influenza strains. Ferrets are therefore considered the best small animal model for predicting IAV disease outcome in humans[31]. Elucidating influenza T cell responses in the ferret model can bridge the gap between murine and human research, thereby facilitating the development of improved influenza vaccines.

Although reagents for the ferret model are still scarce, we recently developed techniques to study (respiratory) T cell responses in IAV infections. Due to this we had the unique opportunity to investigate the role of systemic and respiratory T cells in the protection against H2N2 infection in ferrets. We show that priming with seasonal influenza H1N1 (A/California/ 07/2009) induced a T cell response that could reduce H2N2 (A/ Singapore/1/57) viral replication and disease. H1N1-induced T cells responded to H2N2 and both systemic and respiratory T cell responses were boosted by heterosubtypic H2N2 infection. Respiratory CD8 T cell responses were especially high, even without prior H1N1-priming. Importantly, a group of healthy human donors that was too young to have been into contact with H2N2, displayed responses to H2N2 peptide pools. Together, these results argue that T cells induced by infections with seasonal influenza strains can contribute to protection against pandemic H2N2 infections. Future influenza vaccination strategies should take into account that inducing or boosting T cell responses can protect against heterosubtypic influenza infections.

## Results

### H1N1 immunity reduces H2N2 viral replication and disease.
Infection with circulating IAV might partially protect against H2N2 influenza strains by inducing T cells that recognize shared epitopes (hereafter called "cross-reactive T cells"). To estimate the likelihood of shared epitopes, we first assessed the level of protein identity between IAV subtypes. Viral protein sequences from early human and recent avian H2N2 isolates were retrieved from influenza database GisAid[32] and aligned against recently circulating A/California/07/2009 (hereafter called H1N1). While HA (<65%) and NA (<44%) are poorly conserved, most internal proteins retain >90% sequence identity (Fig. 1a). We then determined how many human CD4 and CD8 T-cell epitopes were conserved between H1N1 and a pandemic H2N2 strain (A/Singapore/1/57; hereafter called H2N2) using the Immune Epitope Database[33]. Corresponding with the sequence identity, epitopes were relatively well conserved for internal IAV proteins, but not for HA and NA (Fig. 1b).

Based on the sequence identity between H1N1 and H2N2, we expect that cellular immunity induced by H1N1 priming can protect against H2N2 infection. To confirm this in a relevant influenza model, we intranasally primed six female ferrets with H1N1 or administered ferrets with PBS as a treatment control (day 0; Fig. 2a and Supplementary Fig. 1). When 4 weeks later both groups were infected with H2N2, animals primed with H1N1 showed decreased viral replication in the throat and nose (Fig. 2b). Correspondingly, weight loss and fever was less severe in primed animals (Fig. 2c, d). A control group that received PBS on both occasions showed baseline weight and temperature and no detectable viral replication. Together, these results confirm that immunity induced by H1N1 priming can protect against heterosubtypic H2N2 infection, as seen by reduced disease symptoms and viral replication.

### H1N1-priming induces cross-reactive T-cell responses.
Next, we examined the role of humoral and cellular responses in the protection against H2N2 infection. H1N1 priming did not raise detectable hemagglutination inhibition (HI) or virus neutralization (VN) titers against H2N2 (Fig. 3a, b). This shows that disease was not reduced by neutralization of virus particles. However, we did detect low levels of binding to recombinant H2-protein by ELISA, indicating that there was cross-reactivity of H1N1-induced antibodies with H2-protein (Supplementary Fig. 2). Importantly, the level of H2-binding antibodies after a single H1N1 infection was relatively low when compared to after H2N2

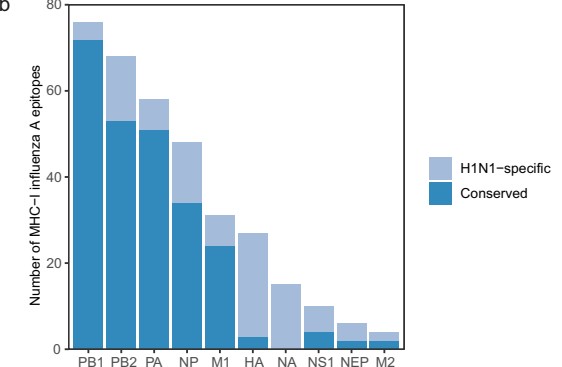

**Fig. 1 Epitopes are conserved between H1N1 and H2N2 influenza A subtypes. a** Sequence identity of influenza A proteins between different H1N1, H2N2, and H3N2 strains. Sequence identity is expressed as percent amino acid overlap with A/California/07/2009 (H1N1). As H1N1 does not express PB1-F2, all PB1-F2 sequences have been excluded from analysis. **b** Influenza epitopes presented by human MHC class-I were mapped against the protein sequence of H1N1 and A/Singapore/1/57 (H2N2). Conserved epitopes (dark blue) are present in both H1N1 and H2N2, while H1N1-specific epitopes (light blue) are only present in H1N1.

infection. When we measured IFNγ-responses in PBMCs by ELISpot prior to H2N2 infection, we observed not only responses to H1N1, but also against H2N2 in primed animals (Fig. 3c). PBMCs from these animals additionally responded against H3N2 influenza (A/Switzerland/97/15293/2013), but not against a nonspecific viral control (mumps), proving the observed responses were influenza-specific. These results are in agreement with prior findings that T cells respond to multiple influenza subtypes by recognizing conserved epitopes[34–36].

To determine the target-proteins of the T-cell response, we stimulated PBMCs with peptide pools spanning the internal proteins of H2N2 and assessed IFNγ-responses by ELISpot. Following H1N1-priming, most responses were observed against NP (59% ± 26) and NS1 (22% ± 22) H2N2 peptide pools (Fig. 3d, e). Virtually no responses against M2, NEP, or PB-F2 were detected. Naïve animals did not respond to any of the H2N2 peptide pools (Supplementary Fig. 3). The high variability in responses between animals is likely in part due to MHC diversity in outbred ferrets. Despite this variation, it is clear that H1N1 priming induced T cells that respond to internal proteins of H2N2.

**H1N1 immunity results in higher CD8 T-cell numbers in the airways upon heterosubtypic infection.** Previous studies have shown that T cells residing in the respiratory tract are essential for the protection against heterosubtypic influenza infections[20,21]. To assess whether respiratory T cells contributed to protection against H2N2 infection, we first determined absolute T-cell numbers in the bronchoalveolar lavage (BAL) two weeks after H2N2 infection. Using a non-centrifugation staining protocol for flow cytometry, we found no difference in absolute CD4 T-cell counts (Fig. 4a). However, CD8 T-cell counts were significantly increased after H2N2 infection with the largest increase seen in primed animals (Fig. 4a, b).

We also investigated T-cell numbers in nasal turbinates (NT), but due to the large variation in absolute cell counts that is inherent with creating a single-cell suspension from tissue, we could only analyze relative T-cell numbers for NT. Similar to BAL, the NT of placebo animals contained few CD8 T cells (~3% of lymphocytes) and relative numbers were increased after H2N2 infection in non-primed ferrets (~12.5%; Fig. 4c). Again, we observed that the response in primed animals was substantially higher (~29.7%). Relative CD4 T-cell numbers remained similar between groups (Fig. 4a).

Infiltrating T cells might be attracted due to inflammation without being responsive to IAV. By plotting the number of infiltrating CD8 T cells versus the number of IAV responsive cells determined by IFNγ-ELISpot, we can derive whether those infiltrating T cells are also IAV-responsive. As expected, increased CD8 T-cell numbers in the BAL correlated with higher IFNγ-responses against IAV (Fig. 4d). Importantly, responses against H1N1, H2N2, and H3N2 all correlated with CD8 T cell influx, demonstrating that the infiltrating CD8 T-cell population recognizes a broad range of IAV subtypes. Although we did not investigate the presence of T cells in the respiratory tract of H1N1-primed animals before H2N2 infection, we did find that a single influenza infection (with H2N2; Fig. 4b) is sufficient to induce respiratory T cells. Furthermore, respiratory T cell numbers were increased in H1N1-primed ferrets after H2N2 infection. This suggests that these T cells were at least in part induced by primary H1N1 infection, and that they responded to secondary H2N2 infection and likely contributed to reduced disease severity.

**T-cell responses in the lung are high but not boosted by heterosubtypic infection.** To further dissect the T-cell response in lung, we measured IFNγ-production in CD4 and CD8 T cells by flow cytometry after ex vivo stimulation with IAV. As lung tissue is rich in vasculature, we perfused the lungs with saline solution

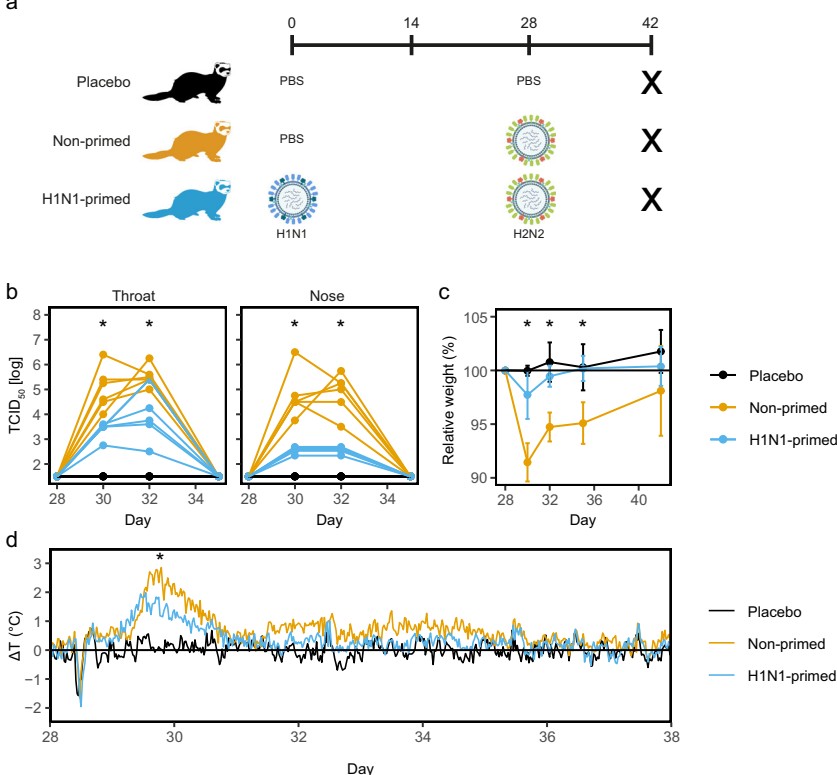

**Fig. 2 H2N2 viral replication and disease manifestation are reduced by prior H1N1-priming. a** Outline of the study. Ferrets were primed with H1N1 (A/California/07/2009) or received PBS on day 0. Subsequently, both primed and non-primed animals were infected with H2N2 (A/Singapore/1/57) on day 28. The placebo group received PBS on both occasions. All animals were euthanized on day 42 to analyze T cell responses. **b** Viral replication in throat and nose of H2N2-infected ferrets prior to infection (day 28) and 2, 4, and 7 days after infection (days 30, 32, and 35 respectively). Lines depict individual ferrets ($n = 6$). **c** Relative changes in weight from the moment of H2N2 infection until the end of the experiment ($n = 6$). Data are displayed as mean ± SD. **d** Temperature displayed as mean deviation from baseline in 30 min intervals from the day of H2N2 infection until day 38 ($n = 5$–6). Baseline temperature was calculated as the average temperature over 4 days prior to H2N2 infection. Asterisk indicates significant differences ($p < 0.05$) between non-primed and H1N1-primed groups after correction for multiple testing. **a** was created using Biorender.

to reduce contamination by blood lymphocytes. After stimulation with H1N1, H2N2, or H3N2, CD8 T cells from infected animals responded strongly with up to 40% of all CD8 T cells being positive for IFNγ (Fig. 5a, b). Interestingly, responses were similar between primed- and non-primed animals. As CD8 T-cell responses in the lung are high even after primary infection, further boosting of the response might be inhibited to prevent excessive immunopathology. In comparison to the CD8 T-cell compartment, CD4 T-cell IFNγ-responses were smaller but noteworthy, with up to 10% of lung-derived CD4 T cells producing IFNγ. As virus stimulations induced some nonspecific IFNγ-responses in placebo animals, we also stimulated lung-derived T cells with a cocktail of peptide pools spanning NP, PA, and NS1 proteins of H2N2 that elicited high responses in ELISpot assays (Fig. 3d). Stimulation with this peptide cocktail induced a clean T-cell response with hardly any responses in placebo animals (Fig. 5c). The majority of responses was seen in the CD8 T-cell compartment, with relatively few CD4 T-cell responders.

In order to compare T cell responses between respiratory and systemic tissues, we additionally measured T-cell responses in blood and spleen. In contrast to the lung, average CD8 T-cell responses after viral restimulation were clearly increased in the spleen of H1N1-primed animals compared to non-primed animals (Fig. 5d). Similar findings were seen for CD4 and CD8 T cells in the blood, but results were not significant due to the small group size ($n = 3$). Additionally, after stimulation with the H2N2 peptide cocktail, H1N1-primed animals displayed significantly increased CD4 and CD8 T-cell responses in the spleen,

with a similar trend for CD8 T cells in the blood (Fig. 5e). CD4 T cells hardly responded to peptide cocktail stimulation by producing IFNγ. Importantly, CD4 T cells might respond by producing other cytokines that we were unable to measure. Hence, we are likely underestimating the true contribution of the CD4 T-cell compartment. These results suggest that a boost shortly after priming can improve systemic T-cell responses, but hardly leads to increased T-cells responses in lung tissue. Interestingly, CD8 T-cells numbers were increased in the BAL and NT after heterosubtypic infection, indicating that there might be differences between the different sites of the respiratory system.

**Pre-existing H1N1 immunity slightly alters the immune hierarchy of an H2N2 infection.** As some internal IAV proteins are more conserved than others (Fig. 1a, b) we wondered whether priming could affect the response to individual H2N2 proteins after an H2N2 challenge. To test this, we measured the IFNγ-responses of primed and non-primed animals to IAV and H2N2 peptide pools by ELISpot. In agreement with our flow cytometric analysis, responses toward IAV stimulations were higher in the blood of primed animals, but not in the lungs (Fig. 6a). However, when we investigated responses to individual influenza proteins, differences between primed and non-primed animals were marginal (Fig. 6b). Out of all H2N2 peptide pools tested, only responses toward NS1 and possibly NP (n.s.) were higher in H1N1-primed animals. Interestingly, primed animals actually

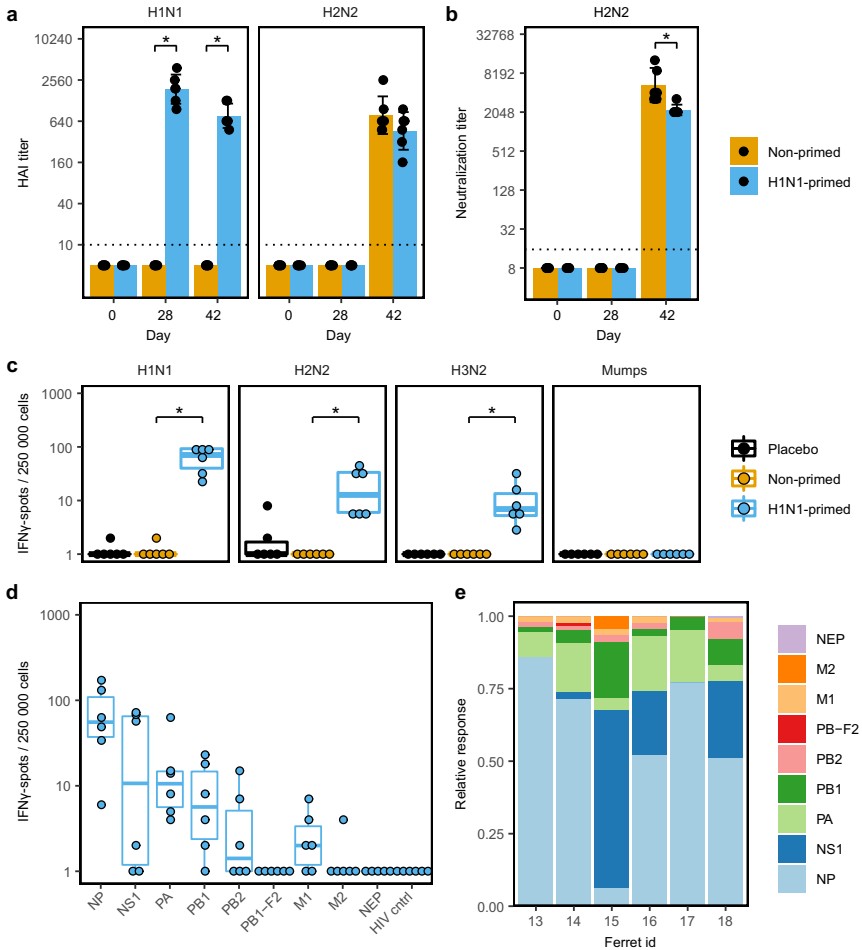

**Fig. 3 H1N1-priming induced H2N2-reactive T cells but no H2N2-neutralizing antibodies. a, b** Sera from primed and non-primed ferrets was tested for antibody responses against A/California/07/2009 (H1N1) and A/Singapore/1/57 (H2N2) before H1N1-priming (day 0), before H2N2 infection (day 28) and 2 weeks after H2N2 infection (day 42). Bars depict mean ± SD. **a** Hemagglutination-inhibition (HAI) titers against H1N1 and H2N2. **b** Virus neutralization titer against H2N2. **c–e** IFNγ-responses by ELISpot after ex vivo stimulation of PBMCs obtained 28 days after H1N1-priming. Responses are corrected for background signals (minus medium stimulation). **c** Responses against H1N1, H2N2, A/Switzerland/97/2013 (H3N2), and mumps virus. **d** Responses of H1N1-primed animals to peptide pools spanning the internal proteins of H2N2 and the control HIV gag protein. Non-primed and placebo animals were negative for all peptide pools tested but were left out for visualization purposes. **e** For each ferret shown in **d**, relative responses against the different influenza proteins was calculated. Boxplots depict the 25, 50, and 75% quantile where the upper and lower whiskers extend to the smallest and largest value respectively within 1.5* the inter quartile ranges. In all plots n = 6. * indicates significant differences (p < 0.05) between non-primed and H1N1-primed groups after correction for multiple testing.

responded less well to PB1 and M2 peptide pools compared to non-primed animals. Although PB1-responses detected after H1N1-priming were boosted by H2N2 infection (Supplementary Fig. 3), the boost was minor and did not exceed the primary PB1-response seen upon a single H2N2 infection. H1N1-priming did not raise any responses to H2N2 M2 peptide pool (Fig. 3d and Supplementary Fig. 3) and while H2N2 infection induced responses to M2 peptide pool in non-primed animals, this was not the case for primed animals (Fig. 6b and Supplementary Fig. 3). Although most responses toward H2N2 peptide pools are similar between primed and non-primed animals, these results suggest that T cell responses can be skewed toward certain antigens by previous infections.

**H2N2 cross-reactive T cells are present in human blood donors**. We have shown that H1N1 priming can induce H2N2 cross-reactive T cells in an animal model. However, this is no guarantee that similar cross-reactivity exists in humans. To investigate whether cross-reactivity to H2N2 is also present in the human population, we measured IFNγ-responses to H2N2

peptide pools in PBMCs of 18 healthy donors (8 male, 10 female). All donors were born after 1968, when the H2N2 influenza subtype was no longer circulating. Strikingly, almost all donors responded to stimulations with NP, M1, and PB1 peptide pools, with some individuals additionally reacting to PA, PB2 and NS1 (Fig. 7). These findings show that H2N2 cross-reactive T cells are present in the human population. Based on earlier studies and our findings in the ferret model, these T cells have the potential to reduce H2N2 spread and disease, although their numbers may be too low in the general population. Boosting the responses by vaccination could therefore be a strategy to limit the consequences of newly introduced influenza subtypes such as H2N2.

## Discussion
With this study we confirmed prior murine work in a more relevant model for influenza disease. We present evidence that (respiratory) T cells can protect against heterosubtypic influenza infection in the ferret model. Priming with H1N1 lead to reduced viral replication and disease upon H2N2 infection, which was associated with the presence of cross-reactive influenza-specific

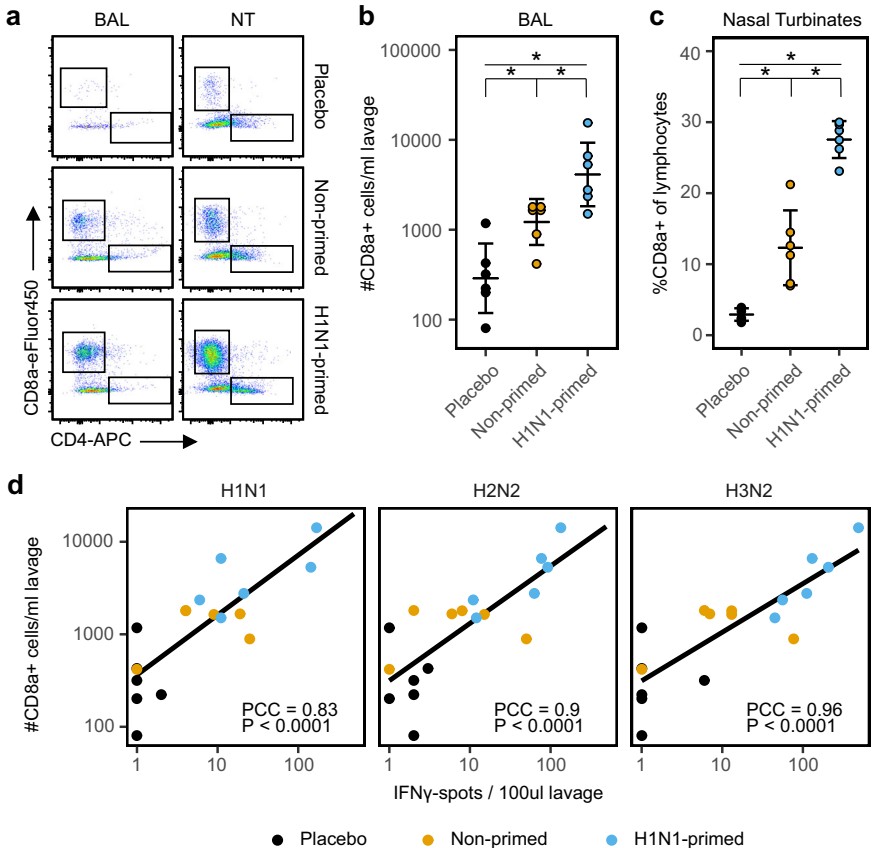

**Fig. 4 Pre-existing T-cell immunity leads to increased infiltration of CD8 T cells into the respiratory tract upon heterologous influenza infection.**
**a** Representative flow cytometry plots depicting the infiltration of CD4 and CD8 T cells in the BAL and nasal turbinates in ferrets 2 weeks after H2N2 infection (day 42). Cells were gated on lymphocytes; singlets; CD3+ and plotted as CD4-APC vs CD8a-eFluor450. **b** Summary of absolute CD8 T-cell numbers in the bronchoalveolar lavage (BAL) after H2N2 infection, displayed as geometric mean ± relative SD. **c** Relative CD8 T-cell number in the nasal turbinates 2 weeks after H2N2 infection. Percentages were calculated as number of CD8a+ cells within all lymphocyte singlets and are displayed as mean ± SD. **d** Correlation between influenza-responsive lymphocytes and the influx of CD8 T cells in BAL on day 42. Absolute CD8a+ T-cell numbers and IFNγ-responses toward several virus stimulations were assessed by flow cytometry and ELISpot respectively. The PCC indicates Pearson's correlation coefficient between spot count and number of CD8a+ T cells in BAL. *$p < 0.05$ after correction for multiple testing.

T cells in the blood before H2N2 infection. Correspondingly, H2N2 infection boosted the T-cell responses that were evoked by H1N1-priming. This was illustrated by increased CD8 T-cell numbers in the respiratory tract of primed animals and higher IFNγ-responses in spleen and blood. Importantly, we could confirm the finding that infection with other influenza subtypes can induce T-cell responses to pandemic H2N2 in healthy human blood donors. By using both the ferret model and human blood donors, we partly mitigated the shortcomings that are associated with murine and human influenza studies.

The Cleveland Family study which ran from 1947 to 1957 reported that adults pre-exposed to H1N1 displayed reduced H2N2 disease[11,12]. With our study we have clearly shown that H1N1 priming induces cross-reactive T cells and that they are associated with protection against H2N2 infection in ferrets. H2N2-responsive T cells are also present in our selection of human blood donors and it is thus likely that T cells played a role in the protection against H2N2 observed in the Cleveland study. This stresses the importance of the T cell response in hetero-subtypic influenza infections. Later studies with influenza-infected individuals extended the initial findings of the Cleveland Family study by showing that high numbers of IAV cross-reactive T cells in the blood correlate with improved clinical outcome in H1N1[13,15], H3N2[15], and H7N9[14] infections. Like the Cleveland study, these studies are however limited by the

availability of tissue (mainly PBMCs) and the unknown infection history of subjects. The more fundamental questions regarding T-cell immunity in influenza infections can therefore only be investigated in animal models.

Earlier ferret studies have shown that prior exposure to IAV can protect against heterosubtypic IAV infection[16–18,37–42]. While some studies especially investigated the role of T cells in this protection, thus far none have addressed the involvement of CD4 and CD8 populations in multiple compartments of the respiratory tract in ferrets. This is especially important as tissue-resident memory T cells (Trm) – a subset of T cells that are non-circulating and respond rapidly upon a (recurring) infection[21,22] – can reduce disease severity and duration upon heterosubtypic infections[20,21,43,44]. In order to study respiratory T cells, we set up a lung perfusion model, which enabled us to analyze a relatively pure population of lung-derived lymphocytes. Additionally, we developed techniques to isolate T cells from the upper respiratory tract (nasal turbinates), which have recently been attributed an important role in blocking dissemination of IAV infection to the lower respiratory tract[23]. These techniques allowed us to investigate the T-cell response in respiratory tissues, although we lacked the reagents to determine whether these cells were also truly tissue-resident.

Unfortunately, we could not investigate T cells in respiratory tissues of H1N1-primed animals before H2N2 infection.

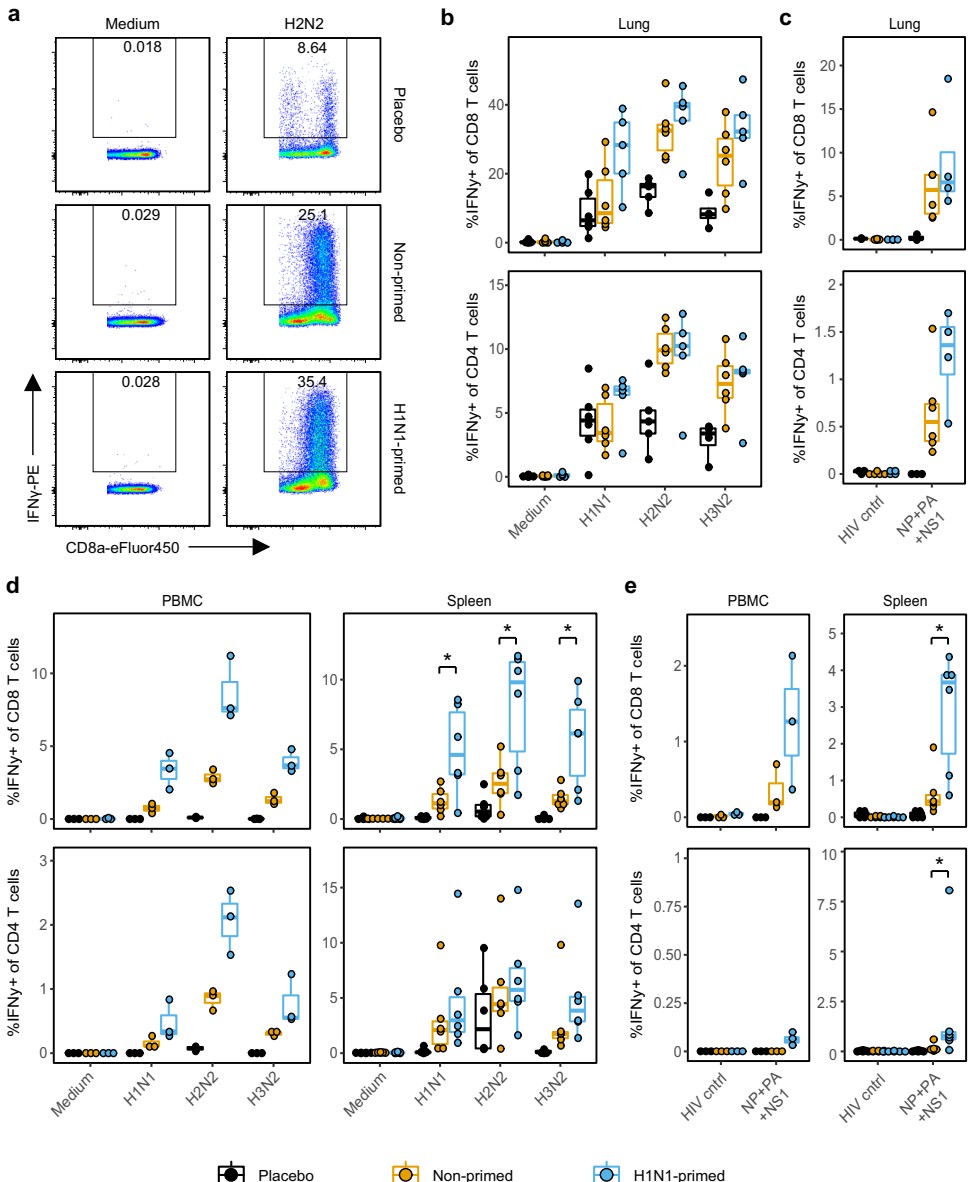

**Fig. 5 IFNγ-responses in the lung are high and only systemic responses are boosted by heterosubtypic H2N2 infection.** Lymphocytes were isolated from blood, spleen, and perfused lungs 14 days after H2N2 infection and stimulated for 6 (peptides) or 24 (viruses) h. IFNγ-responses were then quantified by intracellular cytokine staining. **a** Flow cytometry plot depicting IFNγ-responses in CD8 T cells after medium and H2N2 stimulation of lung lymphocytes. **b**, **c** Percentage IFNγ-producing cells within the CD8 and CD4 T-cell population after stimulation with **b** respective viruses or **c** a peptide cocktail spanning the NP, PA, and NS1 proteins of H2N2 or HIV gag. **d**, **e** Percentage IFNγ producing cells after stimulation of PBMCs or splenocytes with **d** virus or **e** H2N2 peptide cocktail. In panels **b**–**e** $n = 3$ for PBMC, $n = 6$ for spleen and $n = 4$–6 for lung. Boxplots depict the 25, 50, and 75% quantile where the upper and lower whiskers extend to the smallest and largest value respectively within 1.5* the inter quartile ranges. Asterisk indicates significant differences ($p < 0.05$) between non-primed and H1N1-primed groups after correction for multiple testing.

However, as a single H2N2 influenza infection induced a strong T-cell response in the respiratory tract (Fig. 5), it is very likely that this was also the case for the H1N1 infection. Moreover, CD8 T-cell numbers and T cell responses to H2N2 were higher in the respiratory tract of primed animals compared to non-primed animals (Fig. 4), indicating that T cells induced by H1N1 were boosted by H2N2 infection. These findings suggest that influenza-specific T cells were present in respiratory tissues of H1N1-primed ferrets at the time of H2N2 infection and that these cells are associated with reduced H2N2 disease severity.

We also investigated whether humoral immunity could have contributed to reduced H2N2 disease. HAI and VN assays did not

indicate the presence of H2N2-neutralizing antibodies, although it is possible that H1N1-priming induced antibodies which protected against H2N2 by other modes of action. Indeed, in ELISA assays we found that H1N1-induced antibodies could bind to H2N2, albeit at low levels. The antibodies that we detected by ELISA might bind conserved domains in the HA head or stalk, which can mediate protection against heterosubtypic influenza infections[45,46]. However, others have shown that broadly-reactive HA-stalk targeting antibodies attain sufficiently high levels only after repeated vaccination or infection[47].

Although the ferret model is less cost-efficient than mice and is hampered by a lack of reagents, the strong resemblance to human

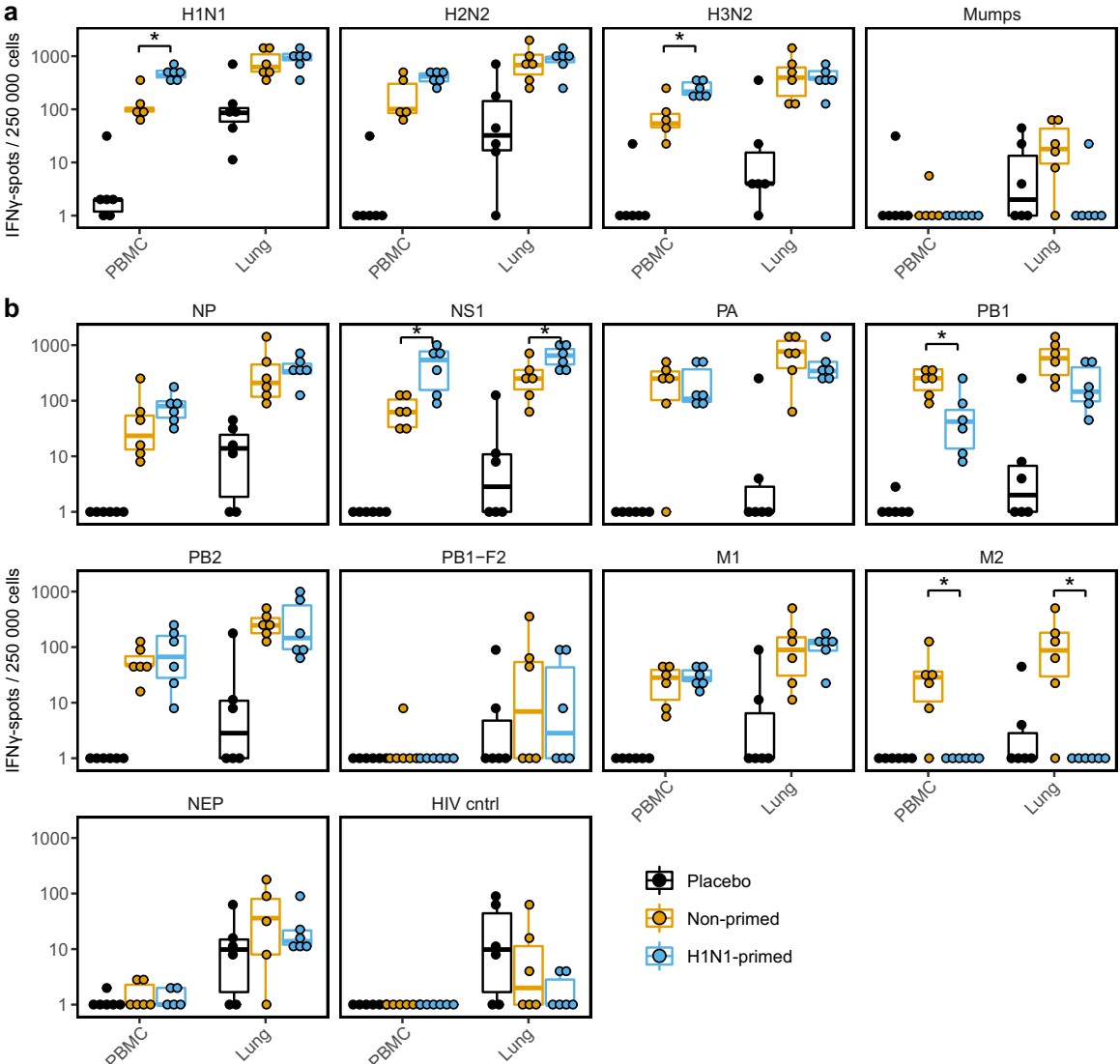

**Fig. 6 Primed animals show increased responses to influenza A virus. a, b** Lymphocytes isolated from blood or lung were stimulated with multiple influenza virus subtypes (**a**) or H2N2 peptide pools (**b**) in an ELISpot assay for 20 h. Responses are corrected for background signals (minus medium stimulation). In all panels, $n = 5–6$. Boxplots depict the 25, 50, and 75% quantile where the upper and lower whiskers extend to the smallest and largest value respectively within 1.5* the inter quartile ranges. Asterisk indicates significant differences ($p < 0.05$) between non-primed and H1N1-primed groups after correction for multiple testing.

influenza infection and disease is a strong motivation to develop the model further for performing studies on correlates of protection. This requires a more thorough understanding of how well the influenza-specific T-cell response in ferrets resembles that of humans. In this study we showed that both H1N1-primed ferrets and healthy human donors display high responses against NP and intermediate responses against PA and NS1 H2N2 peptide pools. Responses against PB-F2, M2, and NEP were absent or very low in both ferrets and humans. M1 responses were high in human samples but low in H1N1-primed ferrets, although H2N2 infection did lead to higher responses (Fig. 6 and Supplementary Fig. 3). This discrepancy might be caused by differences in infection history and/or MHC alleles. Improved understanding of ferret MHC composition and diversity is essential for understanding such similarities and differences between ferret and human T-cell responses.

The current generation of influenza vaccines is focused on inducing HA-neutralizing antibodies. Subunit vaccines contain only HA and NA, while split vaccines additionally contain unknown or small amounts of other influenza A proteins[48,49]. Whole inactivated virus (WIV) vaccines do contain a broader range of influenza proteins. However, inactivated vaccines in general do not infect cells or induce expression of viral proteins, which likely leads to a lower presentation of T-cell epitopes and a reduced T-cell response. With rapid mutations in IAV surface proteins and the threat of pandemics, the use of vaccines that solely focus on humoral responses is an unviable approach. In addition, vaccination of children without prior exposure to influenza might hinder the development of heterosubtypic immunity when traditional inactivated influenza vaccines are used[18,50]. Live attenuated influenza vaccine (LAIV) does infect cells and induce expression of viral proteins[49], but in recent years LAIV vaccination has yielded mixed results (reviewed in Matrajit et al.[51]). Hence, despite these developments we are still in dire need of improved influenza vaccines.

In the end, a successful influenza response requires both the humoral and cellular immune response, in which tissue-residency is an important factor. These branches of the immune system play

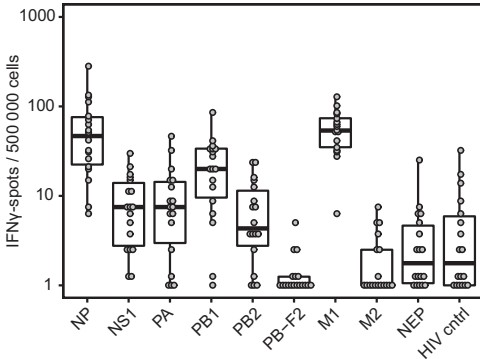

**Fig. 7 H2N2 cross-reactive T cells are present in human blood donors that were born after H2N2 stopped circulating.** Frozen PBMCs from healthy human donors ($n = 18$) were thawed and stimulated with peptide pools coding for the internal proteins of H2N2 or HIV gag in an ELISpot assay. Responses are corrected for background signal (minus medium stimulation). Boxplots depict the 25, 50, and 75% quantile where the upper and lower whiskers extend to the smallest and largest value respectively within 1.5* the inter quartile ranges.

different roles in the response against influenza and complement each other. As the main line of defense, neutralizing antibodies can prevent infection. Cross-reactive antibodies might additionally limit disease upon heterosubtypic infections. If an influenza strain still manages to escapes the humoral immune response, the cellular response can prevent severe disease and reduce viral replication by recognition of conserved epitopes. However, care should be taken to prevent excessive immune activation which could lead to immunopathology[52]. Vaccines that support the interplay of cellular and humoral immunity while preventing excessive immune responses, could help us in reducing future morbidity, mortality, and spread of seasonal and pandemic influenza infections.

## Methods

**Ethical statement.** The experiment was approved by the local Authority for Animal Welfare of the Antonie van Leeuwenhoek terrain (Bilthoven, The Netherlands) under permit number AVD3260020184765 of the Dutch Central Committee for Animal experiments. All procedures were conducted according to EU legislation. Ferrets were examined for general health on a daily basis. If animals showed severe disease according to the defined endpoints prior to scheduled termination they would be euthanized by cardiac bleeding under anesthesia with ketamine (5 mg/kg; Alfasan) and medetomidine (0.1 mg/kg; Orion Pharma). Endpoints were scored based on clinical parameters for activity (0 = active; 1 = active when stimulated; 2 = inactive; and 3 = lethargic) and impaired breathing (0 = normal; 1 = fast breathing; 2 = heavy/stomach breathing). Animals were euthanized when they reached score 3 on activity level (lethargic) or when the combined score of activity and breathing impairment reached 4.

**Viruses.** A/California/07/2009 (H1N1) and A/Switzerland/97-15293/2013 (H3N2) influenza viruses were obtained from the National Institute for Biological Standards and Control (NIBSC, London, England). A/Singapore/1/57 (H2N2) influenza virus was kindly donated by the Institute of Experimental Medicine (IEM, St Petersburg, Russia). All experiments involving H2N2 virus were carried out under BSL-3 conditions. Influenza viruses were grown on MDCK cells in MEM medium (Gibco; Thermo Fisher Scientific) supplemented with 40 μg/ml gentamicin, 0.01 M Tricine and 2 μg/ml TPCK treated trypsin (all from Sigma-Aldrich). At >90% cytopathic effect (CPE), the suspension was collected and spun down (4000 × g for 10 mins) to remove cell debris. H1N1 and H3N2 viruses were sucrose purified on a discontinuous 10–50% sucrose gradient. Due to restrictions inherent with the BSL-3 regime, H2N2 virus suspension was not sucrose purified. Instead, virus suspensions were washed twice on Amicon 100kD Ultra-15 filter units (Merck) with MEM medium. Wild-type mumps virus (MuVi/Utrecht.NLD/40.10; genotype G)[53] was multiplied on Vero cells in DMEM (Gibco) with 2% fetal bovine serum (FBS; HyClone, GE Healthcare). Supernatant of the infected Vero cells was centrifuged at 500 × g and subsequently filtered (5um pore size). All virus aliquots were stored at −80 °C.

**Animal handling.** Animals were housed by subgroup in open cages. From the moment of infection till 14 days after, all groups were housed in BSL-3 level isolators. Animals were visually inspected each day and received food and water ad libitum. For the placement of temperature transponders animals were anesthetized with ketamine (5 mg/kg) and medetomidine (0.1 mg/kg) with 0.2 ml Buprenodale (AST Farma) as a post-operative analgesic. Anesthesia was antagonized with atipamezole (0.25 mg/kg; Orion Pharma). Blood collection from the vena cava on days 0, 14, and 28 happened under similar conditions but without post-operative analgesic. For (mock)-infections, anesthesia likewise consisted of ketamine and medetomidine, but atipamezole administration was delayed by 30 mins to avoid excretion of the inoculum by sneezing and coughing. Weight determinations and swabbing occurred under anesthesia with ketamine alone and did not require an antagonist.

**Study design.** Outbred female ferrets (Schimmel b.v.) aged 18–20 months arrived at the Animal Research Centre (Bilthoven, The Netherlands) at least 3 weeks before commencement of the study for acclimatization. Each treatment/control group consisted of six animals. For practical reasons the experiment was divided into two sub-experiments – named 'A' and 'B' – with each three animals per group. The animals were semi-randomly distributed by weight. Although there were no evident differences between results of the experiments A and B, the statistical analyses used blocking by experiments in order to correct for possible time effects (see 'Statistics' section below).

Animals received temperature transponders (Star Oddi) in the intra peritoneal cavity two weeks before start of the experiment, which recorded body temperature every 30 mins. On day 0, two groups were mock-primed intranasally (i.n.) with PBS ('control' and 'non-primed' groups). A third group was primed with $10^6$ TCID$_{50}$ H1N1 i.n. ('H1N1-primed' group). After four weeks, non-primed and primed groups were infected i.n. with $10^6$ TCID$_{50}$ H2N2 while the control group received a mock-infection i.n. with PBS. For both H1N1 and H2N2 infections, inoculum was administered in 0.1 ml. Prior to infection and on days 2, 4, 7, and 14 after infection, viral nose and throat swabs were collected and animal weight was measured. At the end of the experiment, animals were euthanized by heart puncture and heparin blood and serum was collected. The lungs were then perfused and bronchoalveolar lavage (BAL) was collected by flushing the lungs twice with 30 ml of room temperature RPMI1640 (Gibco, Thermo Fisher). Heparin blood and BAL were used the same day. The spleen, lungs, and nasal turbinates were collected in RPMI1640 and stored overnight at 4 °C. Serum was isolated by centrifugation of clotted blood at 2000 × g for 10 mins and stored at −20 °C until further use.

**Lung perfusion.** Tubing for manual artificial breathing was inserted via an incision in the trachea followed by opening of the chest. A cannula was inserted into the pulmonary artery via an incision in the right ventricle. The abdominal aorta was cut below the junction with the hepatic aorta to allow flushing of the cardiovascular system. Via the cannula the lungs were perfused with physiological saline (B. Braun) until the lungs appeared white and a colorless liquid ran from the abdominal aorta.

**Lymphocyte isolation.** Blood was collected in sodium-heparin coated Vacutainers (BD) and diluted 1:1 with PBS (Gibco) for density centrifugation on a 1:1 mixture of LymphoPrep (1.077 g/ml, Stemcell) and Lympholyte-M (1.0875 g/ml, Cedarlane). Cells were spun down for 30 mins at 800 × g (RT) and the interphase was washed twice using washing medium (RPMI1640 + 1%FBS). The cells were collected in stimulation medium (RPMI1640 + 10% FBS + 1x penicillin-streptomycin-glutamine [Gibco]) and counted using a hemocytometer.

Spleens were homogenized in a sieve using the plunger of a 10-mL syringe after which the suspension was collected in a 50 ml tube. The tube was gently inverted to mix the suspension after which the tube was left for 1–2 mins to let the larger debris sink. The upper 40 ml of cell suspension was transferred to another tube and centrifuged for 5 min at 500×g. The pellet was resuspended in EDTA-supplemented washing medium (RPMI1640 + 1% FBS + 2 mM EDTA (Invitrogen)) and put over a 100 μm cell strainer. The resulting suspension was layered on top of Lympholyte-M and density centrifugation was performed in a similar manner as described for blood. All washing steps were performed with EDTA-supplemented medium to prevent agglutination of cells.

BAL was washed twice with washing medium and resuspended in stimulation medium for further use. Lungs were processed into small cubes of ~5 mm³ and digested in 12 ml of collagenase I (2.4 mg/ml, Merck) and DNase I (1 mg/ml, Novus Biologicals) for 30 mins at 37 °C while rotating. Following this, samples were further homogenized in a sieve using a 10-mL plunger and subsequently washed with EDTA-supplemented washing medium. The suspension was then filtered over a 70-μm cell strainer and used for density centrifugation similar to the spleen.

Nasal turbinates (NT) were gently mashed in a sieve and subsequently filtered over a 70-μm cell strainer. The resulting suspension was rested for 1 min to allow the cartilage fragments to sediment. The suspension – excluding the sedimented cartilage fragments – was transferred to another tube and washed twice with EDTA-supplemented washing medium. The cells were then resuspended in 40% Percoll (GE Healthcare) and layered on top of 70% Percoll. Samples were

centrifuged for 20 mins at $500 \times g$ after which the interphase was collected and washed twice with EDTA-supplemented washing medium. Lymphocytes were resuspended in stimulation medium for further use.

**H2N2 peptide pools**. PepMix[TM] peptide pools for T-cell stimulation assays were obtained from JPT Peptide Technologies GmbH. Each pool contained 15 amino acids long peptides with an overlap of 11 amino acids spanning an entire protein of the internal influenza proteins of the H2N2 A/Leningrad/134/17/1957 strain. The sequence identity between A/Leningrad/134/17/1957 and A/Singapore/1/57 is ≥98% for all proteins, excluding NA and HA. The peptides were synthesized by FMOC-chemistry using a peptide synthesizer and analyzed by LC-MS. Before use, freeze-dried peptide pools were dissolved in DMSO, aliquoted and stored at −20 °C. On the day of use, peptide pool aliquots were thawed and diluted with stimulation medium. The peptide pool suspension was added to cells such that a final peptide concentration of 1 µg/ml per peptide with a DMSO concentration of <0.2% was achieved.

**IFNγ responses by flow cytometry**. In total, 1–3 million lymphocytes were stimulated with virus at MOI 1 for 24 h or H2N2 peptide pools for 8 h. For the last 6 hours of stimulation, Brefeldin A (Golgiplug, BD) was added to the cells, followed by storage o/n at 4 °C. The next day, cells were washed twice with FACS buffer (2 mM EDTA, 0.5% BSA in PBS) and extracellular staining was performed in 100 µl FACS buffer with live-dead aqua (Invitrogen), α-CD4-APC (02, Sino Biological) and α-CD8a-eFluor450 (OKT8, eBioscience) for 30 mins at 4 °C. After washing, cells were fixated and permeabilised by the Foxp3/Transcription factor staining buffer set (eBioscience) according to the manufacturers protocol. Cells were then stained intracellularly with α-CD3e-FITC (CD3–12, Biorad), α-CD79a-APC/eFluor780 (eBioscience), and α-IFNγ-PE (CC302, Bioconnect) for 30 mins at 4 °C. After washing twice, the pellet was resuspended in FACS buffer and measured on a LSR Fortessa X-20 (BD). Data were analyzed using FlowJo[TM] Software V10 (BD) and an example of the gating strategy is presented in Supplementary Fig. 4a.

**Cell counts by flow cytometry**. To reduce cell loss inherent to washing and centrifugation steps during staining, NT and BAL samples were stained using the non-centrifugation PerFix-NC kit (Beckman Coulter) according to the manufacturers protocol. In brief, cells were stained with α-CD4-APC, α-CD8a-eFluor450, and α-CD14-PE (Tük4; Thermo Fisher) for 30 mins at RT. Subsequently, cells were fixated with 25 µl Fixative Reagent for 15 min followed by permeabilization by the addition of 300 µl of Permeabilizing Reagent containing α-CD3e-FITC and α-CD79a-APC/eFluor780. Cells were intracellularly stained for 30 mins at RT, after which 3 ml of Final Reagent was added to each tube. To concentrate the cells the tube was spun down ($500 \times g$, 5 min) and 2.5 ml of the liquid was discarded while the pellet was resuspended in the remaining volume. In total, 50 µl of Coulter Flow-Count Fluorospheres (Beckman Coulter) was added to each sample and the sample was vortexed just before measurement on a LSR Fortessa X-20. Data was analyzed using FlowJo[TM] Software V10 (BD) and an example of the gating strategy is presented in Supplementary Fig. 4b.

**ELISpot**. Pre-coated Ferret IFNγ-ELISpot (ALP) plates (Mabtech) were used according to the manufacturers protocol. Lymphocytes were stimulated with live virus (MOI 1 or 0.1) or H2N2 peptide pools in ELISpot plates at 37 °C. Per well, 250 K cells (PBMC, splenocytes), 125 K cells (nasal turbinates), or 62.5 K cells (lung lymphocytes) or undiluted cell suspension (BAL) was added. After 20 h the plates were developed according to the manufacturers protocol, with the modification that the first antibody staining was performed overnight at 4 °C. Plates were left to dry for 2–3 days after which they were packaged under BSL-3 conditions and heated to 65 °C for 3 h to inactivate any remaining infectious influenza particles. Analysis of ELISpot plates was performed using the ImmunoSpot® S6 CORE (CTL, Cleveland, OH).

**Virus titer analysis**. Nose and throat swabs were collected in 2 ml transport medium containing 15% sucrose (Merck), 2.5 µg/ml Amphotericin B, 100 U/ml penicillin, 100 µg/ml streptomycin, and 250 µg/ml gentamicin (all from Sigma) and stored at −80 °C. For analysis, swabs were thawed, vortexed, serially diluted, and tested in 6-plo on MDCK cells. CPE was scored after 5 days of culturing and $TCID_{50}$ values were calculated using the Reed & Muench method.

**ELISA**. Immulon 2 HB 96-well plates (Thermo Fisher) were coated overnight at RT with 100 µl/well recombinant H2-protein of A/Singapore/1/57 (0.5 µg/ml; BEI Resources; NR-2668). Sera were 2-fold serially diluted in PBS supplemented with 0.1% Tween-80, incubated on the coated plates for 60 mins at 37 °C, and subsequently washed thrice with 0.1% Tween-80. HRP-conjugated goat anti-ferret IgG (Alpha Diagnostic) was diluted 1:5000 in PBS containing 0.1% Tween-80 and 0.5% Protivar (Nutricia) and used to stain H2 bound antibodies for 60 mins at 37 °C. Plates were then washed thrice with 0.1% Tween-80 to remove residual antibodies and once with normal PBS to remove Tween-80. Plates were developed for 10 mins with SureBlue™ TMB (KPL) substrate, after which development was stopped by the addition of 100 µl 2 M $H_2SO_4$ per well. $OD_{450}$-values were determined on the EL808 absorbance reader (Bio-Tek Instruments) and individual curves were visualized using local polynomial regression fitting with R software[54].

**Serological analysis**. Hemagglutination inhibition (HAI) titers in ferret sera were determined according to WHO guidelines[55]. In brief, sera were heat-inactivated at 56 °C for 30 min, treated with receptor destroying enzyme (Sigma), and tested in duplicate against four hemagglutinating units of H1N1 or H2N2 using 1% turkey red blood cells (bioTRADING Mijdrecht).

**Human blood donors**. Buffy coats from healthy individuals born after 1968 were obtained from the Dutch blood bank (Sanquin, Netherlands). Donors provided written informed consent and the study was approved by the Medical Ethics Committee of Sanquin Blood Supply. PBMCs were isolated by density centrifugation as described before[56] and stored at −135 °C till further use. Samples were thawed, resuspended in stimulation medium, and rested for 3 h at 37 °C. ELISpot plates were coated with the Human IFN-γ ELISpot[BASIC] kit (Mabtech) and stimulations were performed as described above with 400K PBMCs per stimulus.

**Influenza A sequence and epitope analysis**. Protein sequences of influenza A subtypes were retrieved from the GISAID database[32] (www.gisaid.org [accessed 9-4-2019]) and aligned against A/California/07/2009 using the online NIH protein blast tool. For the epitope analysis, influenza A epitopes were retrieved from the Immune Epitope Database and Analysis Resource[33] (IEDB; www.iedb.org [accessed on 25-9-2019]). A search for linear, MHC class-I restricted and assay-confirmed epitopes yielded a total of 849 epitopes for all influenza A proteins. After exclusion of non-unique epitopes and alignment against A/California/07/2009 using R software[54], a total of 343 epitopes was found to be present in proteins of A/California/07/2009. These 343 epitopes were subsequently mapped against the proteins of A/Singapore/1/57 to determine if these epitopes were conserved between strains.

**Statistics and reproducibility**. The experiment was divided into two subgroups, A and B, for practical reasons. Although we did not see clear differences between experiments A and B, we included the experimental subgroup as a blocking factor in our analysis. In brief, differences between groups were analyzed using R software[54] and images were visualized with the R package ggplot2[57]. Treatment groups, namely H1N1-primed and Non-primed, were compared by means of the permutation test for differences in group averages with experiment (time) as block implemented in the R package coin[58], with p-values estimated by 10,000,000 simulations. Assays analyzed by this method include $TCID_{50}$ determinations on MDCK cells sampled from the throat and from the nose at days 30 and 32, and with respect to weight at days 30, 32, 35, and 42. The same test was used to compare the two treatment groups and each treatment group with the placebo group with respect to the following endpoints: maximum temperature between days 28–38, infiltration of CD8 T cells in the BAL and NT, percent IFNγ+ cells within CD4 and in CD8 T-cell population sampled from spleen, lungs and PBMC, IFNγ-spot counts for lungs and PBMC. The results of the tests were corrected for multiple testing by using the Benjamini–Hochberg method[59] at a nominal false discovery rate (FDR) of 10%. Only the results that passed the correction have been reported as findings in the results section, which, roughly speaking, means that at most 10% of our findings are likely to be spurious. Associations between IFNγ-spot count and CD8 T cell numbers were tested by using Pearson's correlation coefficient.

**Reporting summary**. Further information on research design is available in the Nature Research Reporting Summary linked to this article.

## Data availability

Results from the statistical analysis are available as Supplementary Data 1. All data supporting the main figures are available as Supplementary Data 2. This includes the GISAID identifiers for influenza sequences and all known influenza A epitopes from the IEDB at the day of retrieval. Note that data presented in Supplementary Data 2 has not been corrected for background responses (Elispot) or log-transformed (Elispot, HAI, and VN titers). All raw data files are being stored inhouse on backed-up servers and are available upon reasonable request to the corresponding author.

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

## Acknowledgements

We would like to thank the biotechnicians of the animal facility for handling of the animals. We are also grateful to prof. van Baarle and dr. Luytjes for critical reviewing of the manuscript.

## Author contributions

K.v.d.V., H.v.D., and F.d.H. performed experiments. K.v.d.V. and J.A.F. analyzed the data. K.v.d.V. and J.d.J. designed the experiment and wrote the manuscript together with J.A.F.

## Competing interests

The authors declare no competing interests.
