## [Peer Review File · Communications Biology]

Reviewers' comments:

Reviewer #1 (Remarks to the Author):

Comments to the authors

In this manuscript van de Ven et al., discuss how the respiratory T cells induced by priming with H1N1 infection protect from H2N2. The concept is novel and indeed relevant with the threat of re-emergence of H2N2 in the community leading to a potential pandemic. The manuscript is well written in general and the experiments support the conclusions. The experimental strategy is good and the experiments plus data analysis are performed neatly. However there are a certain major and minor points which need to be addressed in order to make the manuscript even better.

Major comments

1. The main concern is about the conclusion mentioned in the discussion line 235, "Priming with H1N1 lead to reduced viral replication and disease upon H2N2 infection, which was associated with the presence of cross-reactive influenza-specific T cells in the blood before H2N2 infection." I am afraid that this was observational and the mechanism of these T cells as the sole correlate is not proven. I believe that the cross protection is multifactorial and not dependent on just one mechanism. What are your thoughts about that? I suggest to discuss it in the discussion.
2. Talking about the humoral immunity, the authors show HAI and NT titers against H1N1 and H2N2 for day 28 and 42. On day 28 there are titers against H1N1 only in primed animals while on day 42 there are titers against H2N2 in both primed and non-primed animals which is to be expected of course. But what I miss is the total IgG (and subtype responses) on day 28 against H2N2 which might if found might indicate the role of non-neutralizing antibodies in the cross-protection as well. An ELISA is an option.
3. Also mucosal IgA is supposed to be cross-reactive. Did authors look into presence of these antibodies in nose and throat?
4. In the result section line 131 authors mention "Following H1N1-priming, most responses were observed against NP (59% ± 26) and NS1 (22% ± 22) H2N2 peptide pools (Fig 3d, e). Virtually no responses against M2, NEP or PB-F2 were detected." Quite surprising considering the fact that M2 specifically M2e is quite conserved across different subtypes. Can authors come up with a reasoning for this?
5. What about effect of vaccination? Seasonal vaccine contains A/California/H1N1 but as a split or a subunit antigen. Do authors think that it can induce these T cells as well or one would rely solely on a pre-infection for the protection in case of H2N2 infection? Something to be discussed as well.

Minor comments

1. Introduction line 59 says "Animal studies provided additional proof that T cells are essential for heterosubtypic immunity." references need to be added for this.
2. Please mention humane endpoint for animal sacrifice in the methods section.
3. Both H1N1 and H2N2 were administered in a dose of 10⁶ TCID₅₀. Were these both sublethal doses? What was the rationale behind choosing these? Especially this titer for the priming virus as it can influence the further responses induced. Further upon priming with the H1N1, were there any clinical symptoms in the animals?
4. In figure 7, IFN γ responses are measured against the internal proteins of H2N2. Did author not consider to stimulate with conserved HA epitopes?

Reviewer #2 (Remarks to the Author):

In this work by van de Ven and colleagues, the authors aimed at investigating the role of respiratory T cells in the protection against H2N2 heterosubtypic infection in a ferret H1N1 pre-immune model. The authors found that in an H1N1 pre-immune setting, a recalled systemic and respiratory T cell response, mainly composed by T CD8 cells, correlated with a reduced viral replication and disease following H2N2 challenge. Additionally, the authors confirmed

the presence of H2N2-reactive T cells in healthy human donors that were born when the H2N2 strains were not circulating in the human population, as a demonstration that H1N1 circulating strains could have induced H2N2 cross-reactive T cells. Notwithstanding the need of further investigations, mainly due to the lack of a larger panel of ferret-specific reagents, this manuscript is of some interest for the scientific community working in the field and underline the need of better investigating the T cell response following influenza vaccination, especially in the ferret model, to design and develop next generation influenza vaccines.

Before considering this manuscript for publication in Communications Biology, the authors should address the following criticisms.

Introduction

- Page 3, line 47: This statement is not totally correct. Cross-neutralizing antibodies recognizing the HA stem region can be present.
- Page 3, lines 53-57: the authors should also consider the works by Gostic and colleagues, investigating the role of pre-immunity in heterosubtypic protection.
- Page 4, the authors should also consider possible drawbacks when boosting a T cell response such as the induction of pro-inflammatory cytokines and possible consequent exacerbations of the disease.

Results

- What would be the magnitude of the T cell response in a H3N2 or another group 2 pre-immunity setting?
- Did the authors include samples from individuals that were born when the H2N2 strains were circulating? It would be informative to compare the magnitude of the T cell response between individuals born during and after the H2N2 circulation.

Discussion

- Page 8, line 240: This statement is not totally correct. Depending when blood was collected, not only seasonal but also H1N1 pandemic influenza strains could have also induced a T cell response against H2N2.
- Page 8, line 248: "in" is missing between "present" and "our"
- In the whole manuscript Elispot should be spelled "ELISpot"
- Page 9, the authors conclude that "future vaccination strategies should consider the benefit of inducing or boosting T cell responses for reducing morbidity, mortality and spread of seasonal and pandemic influenza infections". The authors should better describe which strategies can be adopted to achieve this goal.

Referee expertise:

Referee #1: influenza vaccines and immunology

Referee #2: influenza vaccine design

Reviewers' comments:

Reviewer #1 (Remarks to the Author):

Comments to the authors

In this manuscript van de Ven et al., discuss how the respiratory T cells induced by priming with H1N1 infection protect from H2N2. The concept is novel and indeed relevant with the threat of re-emergence of H2N2 in the community leading to a potential pandemic. The manuscript is well written in general and the experiments support the conclusions. The experimental strategy is good and the experiments plus data analysis are performed neatly. However there are a certain major and minor points which need to be addressed in order to make the manuscript even better.

Major comments

1. The main concern is about the conclusion mentioned in the discussion line 235, "Priming with H1N1 lead to reduced viral replication and disease upon H2N2 infection, which was associated with the presence of cross-reactive influenza-specific T cells in the blood before H2N2 infection." I am afraid that this was observational and the mechanism of these T cells as the sole correlate is not proven. I believe that the cross protection is multifactorial and not dependent on just one mechanism. What are your thoughts about that? I suggest to discuss it in the discussion.

We agree with the reviewer that we have not proven that T cells are the sole correlate of protection, nor do we make that claim. We do see an association between the presence of T cells and reduced disease and viral replication. Additionally, we discuss the possible involvement of cross-reactive antibodies later on the discussion (line 303-307).

By recommendation of the reviewer we did perform ELISAs to detect cross-reactive antibodies (further details below). Based on the reviewers comments and the ELISA results we have made some changes in the discussion to clarify that both T cells and antibodies can mediate cross-reactive responses.

2. Talking about the humoral immunity, the authors show HAI and NT titers against H1N1 and H2N2 for day 28 and 42. On day 28 there are titers against H1N1 only in primed animals while on day 42 there are titers against H2N2 in both primed and non-primed animals which is to be expected of course. But what I miss is the total IgG (and subtype responses) on day 28 against H2N2 which might if found might indicate the role of non-neutralizing antibodies in the cross-protection as well. An ELISA is an option.

Based on the reviewers excellent suggestion, we performed ELISAs on sera from H1N1-primed animals (both pre- and post-H2N2 infection; figure below). Experiments have been incorporated in the materials, results and supplemental figures section. In summary, we do see some level of cross-reactive antibodies after a single H1N1 infection, although they are clearly lower compared to an actual H2N2 infection. Based on these findings we further elaborated on the possible role of antibodies in cross-protection in the discussion section.

3. Also mucosal IgA is supposed to be cross-reactive. Did authors look into presence of these antibodies in nose and throat?

Unfortunately we have not been able to measure IgA in earlier ferret experiments due to technical difficulties. Hence, we did not perform this analysis for this experiment.

4. In the result section line 131 authors mention "Following H1N1-priming, most responses were observed against NP ($59\% \pm 26$) and NS1 ($22\% \pm 22$) H2N2 peptide pools (Fig 3d, e). Virtually no responses against M2, NEP or PB-F2 were detected." Quite surprising considering the fact that M2 specifically M2e is quite conserved across different subtypes. Can authors come up with a reasoning for this?

M2e is indeed relatively well conserved between subtypes. It is however also a relatively small protein – thus containing fewer possible epitopes compared to larger proteins. While several (human) M2e epitopes are known, we do not know the composition of ferret MHC. It is possible that the combination of MHC composition and few M2e epitopes resulted in a lack of responses against M2e in our ferret model.

5. What about effect of vaccination? Seasonal vaccine contains A/California/H1N1 but as a split or a subunit antigen. Do authors think that it can induce these T cells as well or one would rely solely on a pre-infection for the protection in case of H2N2 infection? Something to be discussed as well.

Due to the composition of these vaccines - which are focused on HA content – we would not recommend these vaccines for inducing or boosting T cell responses. These vaccines might boost T cell responses against HA or NA epitopes, but due to the variable content of other internal influenza proteins in the split vaccine and no internal proteins in the subunit vaccine, it is unlikely that conserved epitopes are sufficiently processed and presented to induce strong T cell responses. Live attenuated or whole inactivated virus vaccines would be more suitable. We further elaborate on this in our discussion (line 322-333).

Minor comments

1. Introduction line 59 says “Animal studies provided additional proof that T cells are essential for heterosubtypic immunity.” references need to be added for this.

This sentence is an introduction to the paragraph. The individual animal studies (with references) are discussed in the paragraph following this sentence.

2. Please mention humane endpoint for animal sacrifice in the methods section.

We have incorporated this in the methods section under ‘ethical statement’. Of note to the reviewer: in contrast to mice, weight reduction is often not a clear humane endpoint for ferrets due to their large fat reserves and animal-to-animal variation in weight. Hence, while we did measure weight regularly, humane endpoints were based on visual inspection of clinical symptoms as mentioned in the method section.

3. Both H1N1 and H2N2 were administered in a dose of 10⁶ TCID₅₀. Were these both sublethal doses? What was the rationale behind choosing these? Especially this titer for the priming virus as it can influence the further responses induced. Further upon priming with the H1N1, were there any clinical symptoms in the animals?

Both H1N1 and H2N2 infections were sublethal. H1N1 needed to be sublethal in order to be able to infect these animals with H2N2 at a later timepoint. We also chose for a sublethal H2N2 infection so that we could measure immune parameters in animals that only received H2N2 (non-primed). With a lethal challenge, animals of the non-primed group would have likely succumbed to the infection which would have prevented any immunological comparison at 14 days after H2N2 infection.

4. In figure 7, IFN γ responses are measured against the internal proteins of H2N2. Did author not consider to stimulate with conserved HA epitopes?

We agree that inclusion of HA (and NA) peptide pools would have benefitted our assays. Unfortunately, the H2N2 peptide pool was custom made and does not include HA or NA peptide pools. We could have opted to include HA/NA peptide pools of H1N1 (A/California/07/2009), but that would not allow us to test cross-reactivity as we did for the other peptide pools.

Reviewer #2 (Remarks to the Author):

In this work by van de Ven and colleagues, the authors aimed at investigating the role of respiratory T cells in the protection against H2N2 heterosubtypic infection in a ferret H1N1 pre-immune model.

The authors found that in an H1N1 pre-immune setting, a recalled systemic and respiratory T cell response, mainly composed by T CD8 cells, correlated with a reduced viral replication and disease following H2N2 challenge. Additionally, the authors confirmed the presence of H2N2-reactive T cells in healthy human donors that were born when the H2N2 strains were not circulating in the human population, as a demonstration that H1N1 circulating strains could have induced H2N2 cross-reactive T cells. Notwithstanding the need of further investigations, mainly due to the lack of a larger panel of ferret-specific reagents, this manuscript is of some interest for the scientific community working in the field and underline the need of better investigating the T cell response following influenza vaccination, especially in the ferret model, to design and develop next generation influenza vaccines.

Before considering this manuscript for publication in Communications Biology, the authors should address the following criticisms.

Introduction

- Page 3, line 47: This statement is not totally correct. Cross-neutralizing antibodies recognizing the HA stem region can be present.

Cross-reactive antibodies can indeed be present, but stem antibodies are often not neutralizing. We replaced “therefore” with “likely” in order to make a less bold statement.

- Page 3, lines 53-57: the authors should also consider the works by Gostic and colleagues, investigating the role of pre-immunity in heterosubtypic protection.

While Gostic and colleagues nicely explain the role of imprinting in the influenza immune response, they primarily focus on hemagglutinin. In contrast, this section of the manuscript (page 3) focuses on the T cell aspect. Hence we felt that the work of Gostic *et al.* was more suited for the discussion (line 305).

- Page 4, the authors should also consider possible drawbacks when boosting a T cell response such as the induction of pro-inflammatory cytokines and possible consequent exacerbations of the disease.

This is indeed a valid concern. We decided to elaborate on this in the latter sections of the discussion (line 340-342).

Results

- What would be the magnitude of the T cell response in a H3N2 or another group 2 pre-immunity setting?

Based on the conservation of the internal proteins between H1N1, H2N2 and H3N2, we expect the T cell response to be comparable even when priming with H3N2 (or other group 2 influenza strains). As these groups are based mainly on HA, the distinction between group 1 and 2 influenza strains is not necessarily relevant when discussing internal influenza epitopes.

- Did the authors include samples from individuals that were born when the H2N2 strains were circulating? It would be informative to compare the magnitude of the T cell response between individuals born during and after the H2N2 circulation.

We agree with the reviewer that this is indeed an interesting topic. We did not include donors that were born during or before the H2N2 pandemic, but we plan to do so for a future project.

Discussion

- Page 8, line 240: This statement is not totally correct. Depending when blood was collected, not only seasonal but also H1N1 pandemic influenza strains could have also induced a T cell response against H2N2.

This is indeed correct. All blood from human donors was collected >2016 and hence they likely encountered both pre- and post-pandemic H1N1. To prevent confusion, we have therefore rewritten this to 'other influenza subtypes' (as in: non-H2N2) instead of 'seasonal influenza'.

- Page 8, line 248: "in" is missing between "present" and "our"

Thank you for noticing this error. Corrected the text.

- In the whole manuscript Elispot should be spelled "ELISpot"

We corrected this throughout the manuscript.

- Page 9, the authors conclude that "future vaccination strategies should consider the benefit of inducing or boosting T cell responses for reducing morbidity, mortality and spread of seasonal and pandemic influenza infections". The authors should better describe which strategies can be adopted to achieve this goal.

In our opinion it is important that future influenza vaccine contain a broader range of epitopes that boost T cell responses. We expanded the discussion on this point.

REVIEWERS' COMMENTS:

Reviewer #1 (Remarks to the Author):

The authors addressed all the questions adequately. The manuscript reads well.

Reviewer #2 (Remarks to the Author):

In this revised version the authors addressed my criticisms and concerns. In my opinion the manuscript is now suitable for publication.